# Exploring the pathogenesis and immunological profiles of psoriasis complicated with MASLD

**Shuhui Tan**[1‡]**, Mingyue Liu**[1‡]**, Fei Feng**[1‡]**, Ruicheng Li**[2]**, Rui Tian**[2]**, Zhenhua Nie**[3]*

**1** Tianjin University of Traditional Chinese Medicine, Tianjin, China, **2** Tianjin Academy of Traditional Chinese Medicine Affiliated Hospital, Tianjin, China, **3** Tianjin Medical University, Tianjin, China

‡ ST, ML and FF contributed equally to this work as co-first authors.
* niezhenhua@163.com

## Abstract

### Background

Both psoriasis and metabolic dysfunction-associated steatotic liver disease (MASLD) are immune-mediated chronic inflammatory diseases. Psoriasis manifests itself mainly as skin damage, while MASLD mainly involves the liver promoting liver fibrosis, which has a significant impact on patient health and quality of life. Some clinical studies have shown that there are mutually reinforcing mechanisms between these two diseases, but they are not clearly defined, and this paper aims to further explore their common pathogenesis.

### Methods

Gene expression profiling datasets (GSE30999, GSE48452) and single cell datasets (GSE151177, GSE186328) for psoriasis and MASLD were downloaded from the Gene Expression Omnibus (GEO) database. Common differential gene sets were obtained by gene differential analysis, and then functional enrichment of differential genes was performed to find associated transcription factors and PPI protein network analysis. Single-cell datasets were validated for gene expression and explored for cellular communication, gene set differential analysis and immune infiltration analysis.

### Results

We identified seven common differential genes, all of which were upregulated.The IL-17 pathway, tumor necrosis factor (TNF-α) pathway were shown in strong association with both diseases, and five transcription factors regulating the differential genes were predicted. Two key genes (MMP9, CXCL10) and three key transcription factors (TF) (IRF1, STAT1, NFKB1) were obtained by PPI protein network analysis. Single cell dataset verified the expression of key genes, and combined with gene set differential analysis, immune infiltration revealed that CD4+ T cells, NK cells and macrophages were heavily infiltrated in both diseases. IL-17, IL-1 and cGAS-STING pathways were highly expressed in both diseases, and both diseases share a similar immune microenvironment.

**Data Availability Statement:** The datasets (GSE30999, GSE48452, GSE151177, GSE186328) analyzed during this study are publicly available in the Gene Expression Omnibus database (http://www.ncbi.nlm.nih.gov/geo). The original

contributions presented in the study are included in the article/supplementary material. Further inquiries can be directed to the authors.

**Funding:** The author(s) received no specific funding for this work.

**Competing interests:** The authors have declared that no competing interests exist.

## Conclusions

Our study reveals the common pathogenesis of psoriasis and MASLD from gene expression to immune cell similarities and differences, identifies key genes and regulatory pathways common to both, and elucidates the similarities in the immune microenvironment of both diseases, providing new ideas for subsequent studies on targeted therapy.

## 1. Introduction

The skin disease, also known as psoriasis, has a strong genetic predisposition and immune-related pathogenic characteristics. Psoriasis lesions are well-demarcated round red papules or plaques with dry gray or silvery-white scales produced by hyperkeratosis of the skin, and in some subtypes, pustules. The lesions are usually symmetrically distributed on the scalp, elbows, knees, lumbosacral area and body folds, and may also involve the oral mucosa, tongue or eyelids [1]. Psoriasis reduces the patient's quality of life, increases financial burden, causes psychological stress, and in severe cases, can be combined with other life-threatening complications. The prevalence of *psoriasis* varies by region and population. The reported prevalence of psoriasis in East Asian populations is about 0.11% [2]. However, the prevalence in Europe and America is 1%-2% [3, 4]. Studies in the last decade have shown that metabolic syndrome (MetS) and obesity are strongly associated with psoriasis [2].

Non-alcoholic fatty liver disease (NAFLD) was officially renamed in June 2023 as metabolic dysfunction-associated steatotic liver disease (MASLD). It is defined as patients with hepatic steatosis who have at least one of five concurrent cardiovascular metabolic risk factors (BMI/ waist circumference, blood glucose, blood pressure, triglycerides, and HDL-C) are categorized as having MASLD [5]. MASLD encompasses a spectrum of progressive steatohepatitic liver disease ranging from isolated hepatic steatosis to metabolic dysfunction-associated steatohepatitis with varying degrees of hepatic fibrosis (MASH) and may progress to cirrhosis. In recent years, the global prevalence of MASLD is increasing annually, from 25.3% (1990–2006) to 38.2% (2016–2019), with a nearly 50% increase in global prevalence of MASLD over the past three decades [6]. There are also regional differences in prevalence, with the highest rates in South America and the Middle East (30.5% and 31.8%, respectively) and the lowest in Africa (13.5%) [7].Although some studies have shown that psoriasis increases the prevalence of MASLD, the common pathogenesis between these two diseases has been a point of wide interest and discussion.It is now believed that obesity, insulin resistance, persistent inflammation, and the hepatic-dermal axis that develops between the skin, adipose tissue, and liver play important roles in the mechanism of association between the two diseases [8]. In psoriatic lesions, activation of chronic inflammation leads to infiltration of dendritic cells (DCs), T cells, macrophages and neutrophils, which in turn triggers the secretion of cytokines such as tumor necrosis factor (TNF)-α, IL-1, IL-17, and IL-23 by pro-inflammatory leukocytes. These inflammatory factors exacerbate inflammation at the lesions, releasing more inflammatory factors, and creating a vicious cycle. High expression of the IL-17/IL-23 axis and an imbalance of IL-17/Treg cells are thought to contribute to psoriasis. In the liver, tissue inflammation is mediated by the production of pro-inflammatory cytokines, such as TNF-α, IL-6 and IL-17 [7], by resident hepatocytes and immune cells. Similar findings of increased Th17 cells and imbalance in the Th17 cell/ quiescent Treg cell ratio have been observed in patients with MASLD, indicating that the IL-17 axis and Treg cells also play an important role in MASLD [9]. Although many studies have been conducted to investigate the relationship between psoriasis and

MASLD, the exact mechanisms remain uncertain and need further investigation. In the present study, we used a bioinformatic approach to search for key genes associated with the pathogenesis of psoriasis and MASLD, and to explore the cellular pathways under their control, the role of immune cell and the inflammatory response. To provide new directions and ideas for the individualized treatment of such patients.

## 2. Materials and methods

### 2.1. Data source

We retrieved psoriasis and MASLD related datasets from the Gene Expression Omnibus (GEO) database (http://www.ncbi.nlm.nih.gov/geo), respectively, and screened them for humans based on sorted sample size and sample source, and identified two microarray datasets, GSE30999 and GSE48452, and two single-cell datasets, GSE151177 and GSE186328.

### 2.2. Identification of DEGs

GSE30999 and GSE89632 were derived from two platforms GPL570 and GPL11532, respectively. Raw and platform information of the data was downloaded via 'R-4.2.1', and data information was corrected for background, normalized to the data and transformed log2. Differential gene analysis was performed on each of the two datasets using the 'limma' package to obtain differentially expressed genes (DEGs) in disease versus normal subjects, with statistically defined criteria for differential genes being log2FC>1 and adj. P.val<0.05. Those probes without gene names were excluded, and those corresponding to multiple probe sets were averaged to obtain the respective DEGs, and then the differential genes were intersected to eliminate genes with opposite expressions to get common DEGs.

### 2.3. Functional and pathway enrichment analysis of DEGs and prediction of TFs

The selected DEGs were subjected to functional enrichment analysis of Gene Ontology (GO) and Kyoto Encyclopedia of Genes and Genomes (KEGG) pathways (Gene Ontology is a database established by the Gene Ontology Federation and Kyoto Encyclopedia of Genes and Genomes pathways is a database dedicated to storing information on gene pathways in different species). GO enrichment is the analysis of genes in terms of cell composition (CC), molecular function(MF), and involved biological processes involved (BP), while KEGG focuses more on signaling pathways to determine the role of DEG in various metabolic pathways [10]. The TRRUST database (https://www.grnpedia.org/trrust) was also used to predict the transcription factors that regulate these genes.

### 2.4. PPI network construction and selection of hub genes

The GENEMANIA database (http://genemania.org/) was used to retrieve relevant expressed genes and to construct gene expression networks. PPI protein network analysis is a built-in search tool based on the STRING (https://string-db.org) database for functional network analysis between proteins. The corresponding results were structured using the software "Cytoscape 3.9.1" (https://cytoscape.org), and key functional modules were analyzed using the 'Cytoscape MCODE' plug-in, with the following selection criteria: k-core = 2, degree cutoff = 2, max depth = 100, and node score cut-off = 0.2. Hub genes were selected using the 'cytoHubba' plug-in to select the most valuable combined genes.

## 2.5. Single-cell quality control, dimensions reduction and cell type annotation

Single-cell datasets of GSE151177 and GSE186328 for psoriasis and MASLD were processed using the R package 'Seurat.' The GSE151177 psoriasis dataset is a single-cell sequencing set containing all cells of skin tissue from 17 case groups and six controls, was merged using the 'Harmony' package to eliminate batch effects, and the cells with gene expression PTPRC (CD45) > 1 were taken as the immune cell set, and the new subset was normalized for data, scaled down by PCA (dims = 20), binned by UMAP (resolution = 0.3) and annotated for the binned cells. The MASLD dataset GSE186328 contained CD45$^+$ single-cell data from 3 case groups and three controls. The same harmony integration, data normalization, PCA downscaled (dims = 15), UMAP binning (resolution = 0.5), and binning annotation were performed. Cell proportional composition analysis was performed on the annotated dataset to compare the similarities and differences in immune cell composition between the two diseases and to validate the expression of differential genes in immune cells.

## 2.6. Cell-cell communication analysis and gene set variation analysis

Cell-cell communication analysis was performed on the single-cell datasets using the "Cell-Chat" package to find pathways of high cell-to-cell expression. A 20% cutoff was set. The ligand receptors for each pathway and the weight of contribution they accounted for were calculated. We used the "MMF" package to infer global communication patterns, explore how multiple cell types coordinate together, and explore the role of cell signaling pathways in two diseases. Relevant pathway genes were retrieved using the PathCard database (https://path cards.genecards.org/) to form multiple gene sets, and these gene sets were subjected to gene set variation analysis (GSVA) to determine the variability of these pathways in disease groups versus the control group and to explore the relationship between the two diseases in terms of onset progression.

## 2.7. Immune infiltration analysis

The bioinformatics algorithm using of CIBERSORT (https://cibersortx.stanford.edu/) was used to calculate the infiltration of immune cells This approach assumes that the abundance of immune cells is estimated using a reference set containing 22 subtypes of immune cells (LM22) and 1000 alignments (25). The R package "corrplot" was used to correlate and visualize the 22 types of immune cell infiltration.

# 3. Results

## 3.1. Identification of DEGs

We downloaded two gene microarray data, GSE30999 and GSE48452, from the GEO database in R. Differential gene analysis were performed with the "limma" package and obtained 2229 DEGs were obtained in GSE30999 for the control and psoriasis groups, including 1208 upregulated and 1021 downregulated genes (Fig 1A). In GSE48452, there were 17 (14 up-regulated and three down-regulated) and 21 (17 upregulated and four downregulated) differential genes in the healthy-MASLD group and the healthy obese-MASLD group (Fig 1B and 1C). These two DEGs were intersected separately with those of the psoriasis group (Fig 1D) and then combined to exclude genes with opposite phenotypic responses. Finally, seven common differential genes were identified, of which FMO1 was three of these common differential genes, all of which were upregulated.

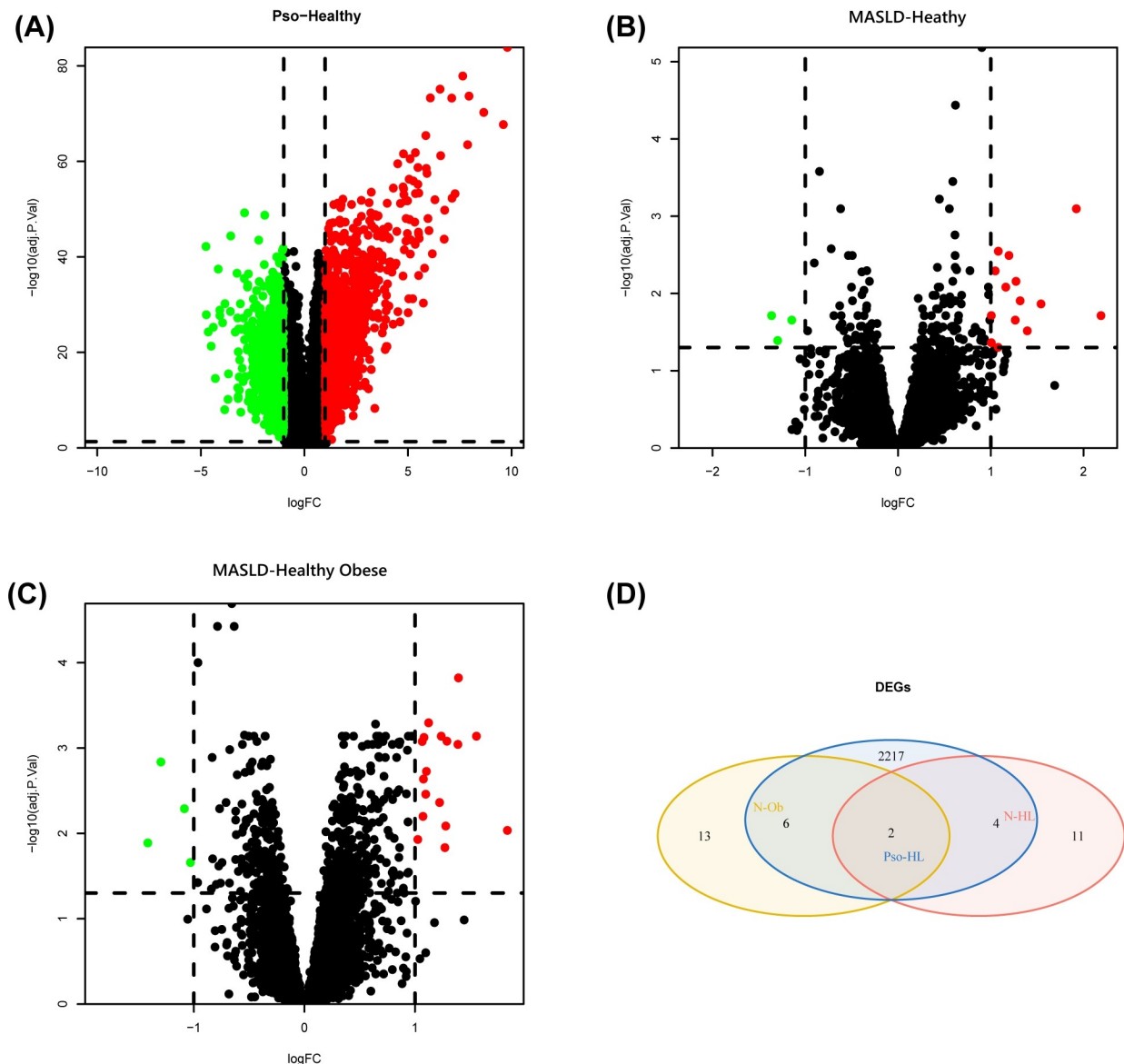

**Fig 1. Volcano diagram and Venn diagram.** (A) The volcano map of GSE30999. (B) (C) The volcano map of GSE28829. Upregulated genes are marked in red; downregulated genes are marked in green. (D) The two sets of differential genes of GSE89632 overlapped with the GSE30999 intersection.

### 3.2. Functional enrichment analysis of DEGs and identification of TFs

GO functional enrichment analysis and KEGG enrichment analysis was performed on the selected DEGs. The results of the GO enrichment analysis showed (Fig 2A), MF: NADP binding. BP: NADP metabolic process, response to vitamin, response to radiation. The results of the enrichment analysis of the KEGG signaling pathway showed that these genes were enriched in the IL-17 and TNF-related pathways with p-values < 0.05 (Fig 2B). TURRST is a transcriptional regulatory network database into which we entered our DEGs and retrieved five corresponding transcription factors IRF1, STAT1, SP1, RELA, and NFKB1 (p<0.05).

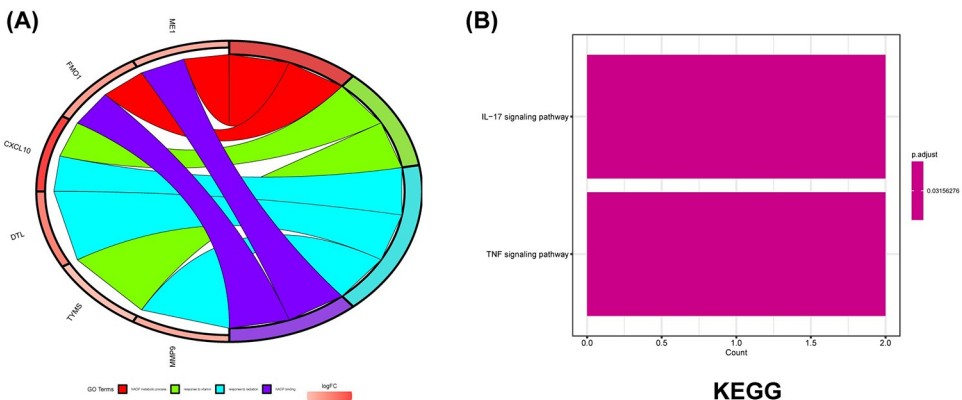

**Fig 2. DEGs enrichment analysis results.** (A) The enrichment analysis results of GO Pathway. (B) The enrichment analysis results of KEGG Pathway.

### 3.3. PPI network construction and selection of hub genes

The co-expression network and associated functions of differential genes and transcription factors were then analyzed according to the GENEMANIA database, where co-expression:46.55%, physical interaction: 35.22%, prediction: 10.51%, pathway: 3.71%, co-localization: 3.05%, shared protein domain: 0.96% (Fig 3A). Twenty related genes were also obtained, and these related functions were mainly involved in regulating inflammatory factors such as interferons and interleukins, the development of inflammation production, and biological processes such as regulating lipid and sterol synthesis. A functional network analysis between proteins was performed for these genes using the built-in search tool of the STRING database. The selection criteria were a composite score of >0.4, excluding those genes that do not interact independently with other genes, and 25 nodes and 96 pairs of interactions were obtained. The obtained PPI analysis was then used to get one tightly connected key gene module, containing ten genes using the MCODE plugin 'cystoscope' (Fig 3B), while the top 5 genes were selected by the built-in intelligent algorithm (DNNC) of the plugin "cytoHubba" (Fig 3C). Five hub genes (IRF1, STAT1, NFKB1, CXCL10, MMP9) were screened, including two previously obtained differential genes and three transcription factors.

### 3.4. Single-cell data processing and cell type annotation

Classification of the GSE151177 psoriasis dataset yielded nine clusters, which we annotated into five cell populations based on common immune cell marker genes: CD4$^+$T cells, CD8$^+$T cells, natural killer cells (NK), macrophages, and DCs (Fig 4A). The MASLD dataset GSE186328 was processed to obtain ten clusters, annotated as CD4$^+$T cells, CD8$^+$T cells, NK cells, and macrophage/ Kupffer cells, neutrophils, and monocyte six cell populations (Fig 4C). Similarly, the proportion of Kupffer cells. cells, monocytes, and neutrophils were higher, and differential genes are more frequently expressed in macrophages (Fig 4B and 4D).

### 3.5. Cell-cell communication analysis

Cell-cell communication was performed for the disease groups in the data and 34 cellular ligand-receptor pairs and their signaling communication were obtained for the psoriasis group and 41 for the MASLD group, involving 26 of the same signaling communication (Fig 5A and 5B). We focus on the communication of some of the more classical inflammatory factors. In terms of emitting and receiving signals, macrophages and dendritic cells contribute

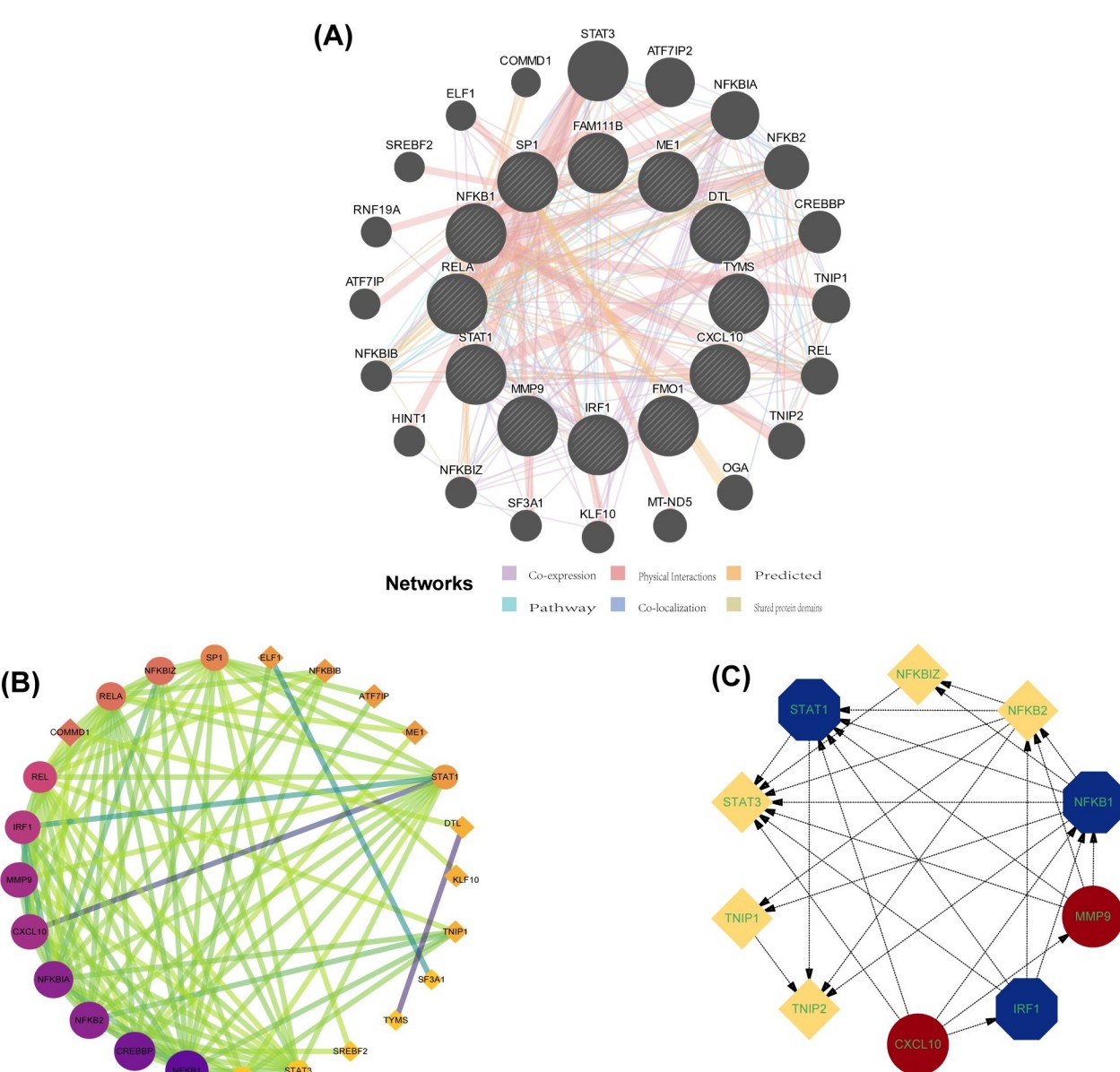

**Fig 3. Gene co-expression network, PPI protein network and key gene modules.** (A) Gene co-expression network with inner circle for differential genes and TF. Outer circle for 20 related co-expressed genes. (B) PPI network. The genes in the circles are the key functional module genes for MCODE analysis. (C) Key functional modules in the screening of hub gene. red represents DEG, blue represents TF.

more to these inflammatory signals in psoriasis and Kupffer cells. monocytes, and neutrophils in MASLD. We found that chemokine signaling had a higher intensity of communication in both diseases, mainly between myeloid and CD4+ cells (Fig 5C–5F). Globally, the high expression of immune cell populations in the myeloid lineage allows one to produce more inflammatory signals (Fig 6A–6D). These signals are mainly involved in the production and delivery of inflammatory factors and belong upstream and downstream of the Jak-Stata pathway, NF-κB pathway, MAPK pathway and PI3-K / AKT pathway, constituting a complex network of signaling pathways. It is highly consistent with our known pathogenic pathways associated with psoriasis, suggesting a complex and similar process of immune cell infiltration and intercellular signaling developing MASLD as in psoriasis. In particular, associated

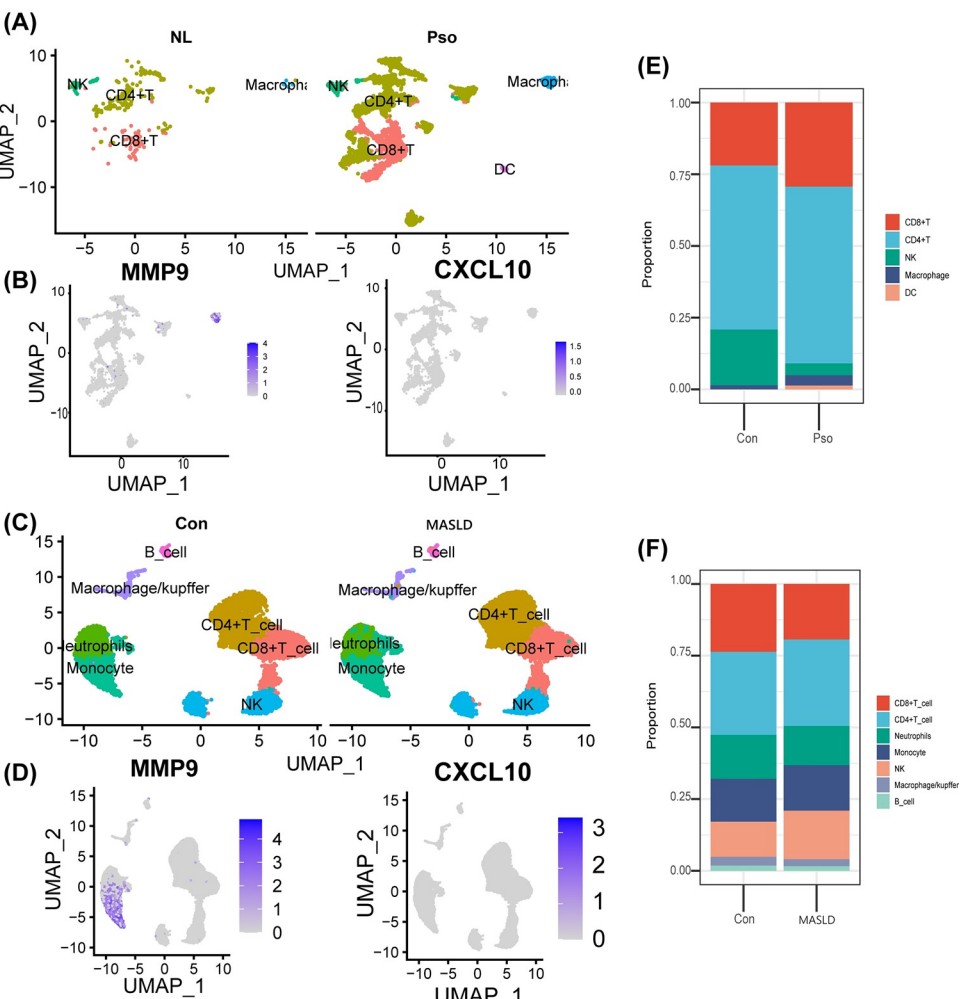

**Fig 4. Single-cell data profiling, DEGs expression and cell proportions.** (A) Dataset of GSE151177 immune cell annotation results, normal vs. psoriasis comparison UMAP plot. (B) Expression plots of MMP9 and CXCL10 among immune cells in psoriasis. (C) Dataset of GSE186328 annotation results, normal vs. MASLD comparison UMAP plot. (D) Expression of MMP9 and CXCL10 in GSE186328. (E) Comparison of immune cell ratio between normal and psoriasis groups in GSE151177. (F) Proportion of immune cells in the normal group versus the MASLD group in GSE186328.

macrophages, CD4$^+$ T cells, play a similar role in inflammation's generation and transmission phases.

## 3.6. Gene set variation analysis

We used the previously obtained relevant and potentially relevant inflammatory pathways mentioned in the literature as key pathways for both disease candidates, and then retrieved the corresponding sets of relevant genes through the PathCard database (S1 Table). GSVA analysis of these gene sets in GSE30999 and GSE48452 revealed that the IL-17 pathway, IL-1 pathway, and cGAS-STING pathway were significantly different in the two diseases (p<0.05) (Fig 6E and 6F). The most prominent function of the cGAS-STING pathway is the production of inflammatory and antiviral proteins in response to microbial nucleic acids, mainly type I interferon (IFN-I) production [11], a pathway that has been associated with both diseases but has previously received little attention.

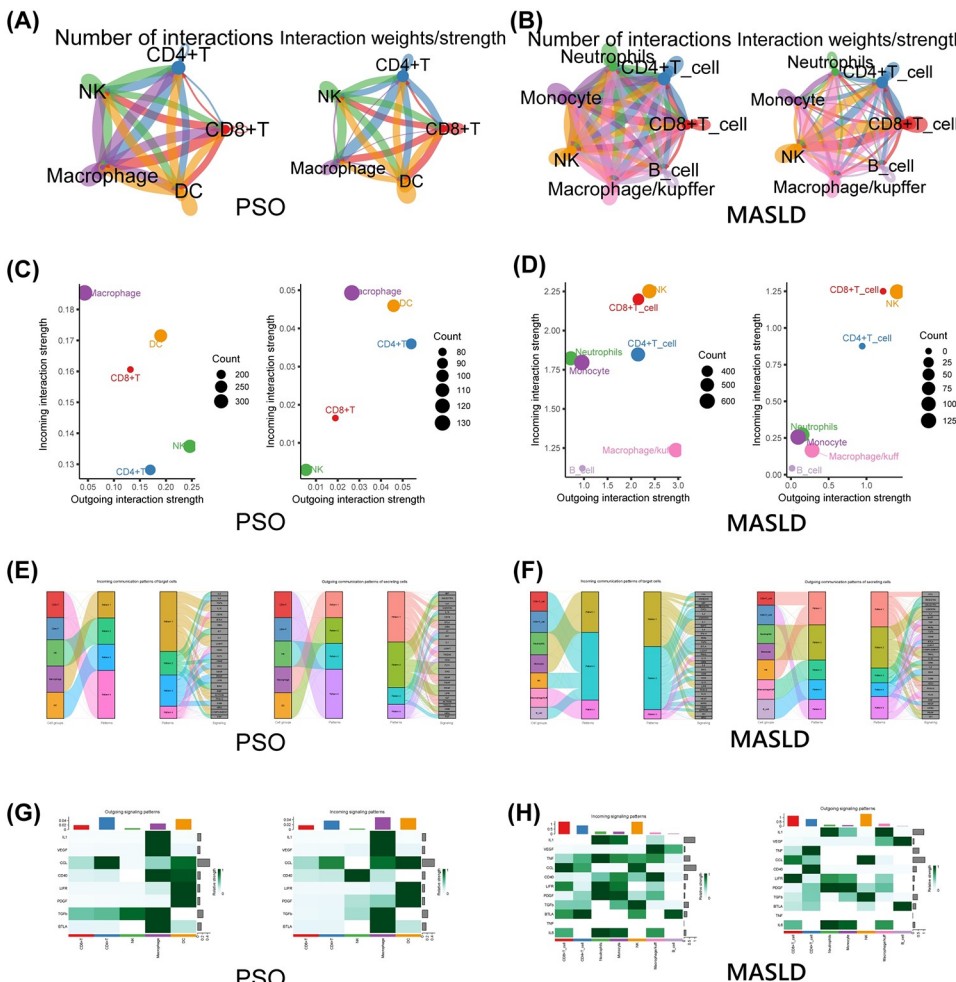

**Fig 5. Visualization of intercellular communication interactions.** (A) (B) Line graphs of the number and intensity of interactions between immune cells in communication in psoriasis and MASLD. (C) (D) 2D dot plots of the intensity of immune cell interactions at the sending and receiving ends of cellular communication in psoriasis and MASLD, with the left and right plots indicating all pathway signals and inflammatory pathway signals, respectively. (E) (F) River plots for clustering cells and signalling pathways into different modules under global analysis for psoriasis and MASLD, respectively. (G) (H) Heat map of immune cells in psoriasis and MASLD with weighted roles in signalling emitted and received in the inflammatory pathway scoring.

## 3.7. Immune infiltration analysis

We calculated the respective immune cell infiltration in psoriasis and MASLD groups according to the CIBERSORT algorithm (Fig 7A and 7B). In both diseases, there was significant infiltration of CD4[+] T cells, NK cells and myeloid cells. In the psoriasis group, monocytes and M1 macrophages were positively correlated with CD4[+]T cells and NK cells. Plasma cells and quiescent DCs were negatively correlated with CD4[+] T cells. In the MASLD group, monocytes and M1 macrophages positively correlated with CD4[+]T cells, quiescent NK cells and activated DCs. Monocytes were negatively correlated with B cells, plasma cells and activated NK cells. Treg cells were negatively correlated with CD4[+] T cells and positively correlated with M0 and M2 macrophages (Fig 7C and 7D).

## 4. Discussion

Early studies, identified TNF-α is a key trigger of the innate inflammatory pathway in psoriasis [12], and as the understanding of the disease has advanced, the IL-23/IL-17A axis at the core

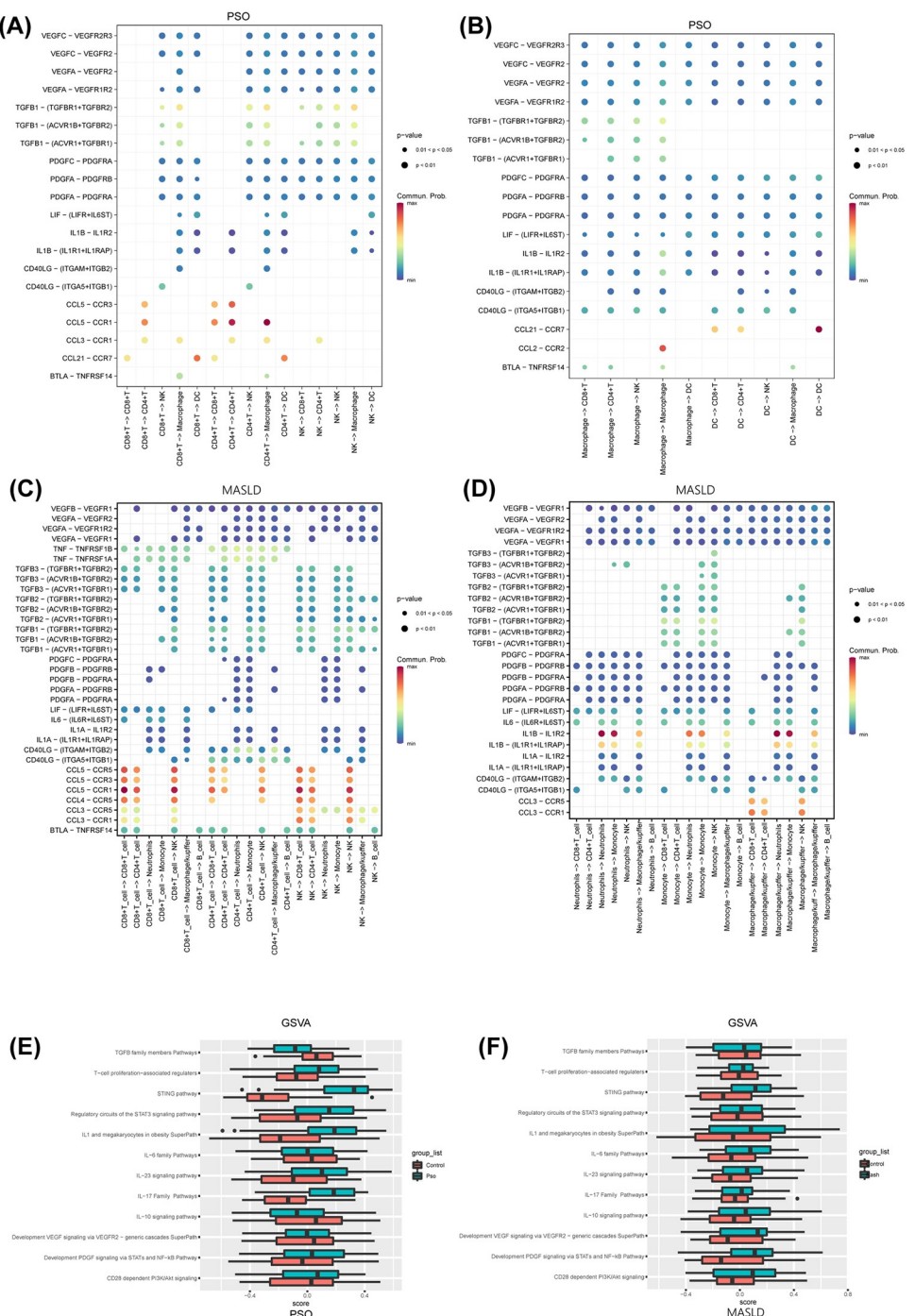

**Fig 6. Significant ligand-receptor dot plots between cell groups and box line plots for GSVA analysis of common pathways.** (A) Intensity of action of lymphatic lineage cells in psoriasis as the emitting end at each ligand receptor. (B) Intensity of action of myeloid cells in psoriasis as the emitting end at each ligand receptor. (C) Intensity of action of lymphatic lineage cells as emitters at individual ligand receptors in MASLD. (D) Intensity of action of myeloid cells as emitter at individual ligand receptors in MASLD. (E) (F) Box plot of GSVA analysis for both diseases, with the horizontal axis showing the expression result scores and the vertical axis showing the names of each pathway.

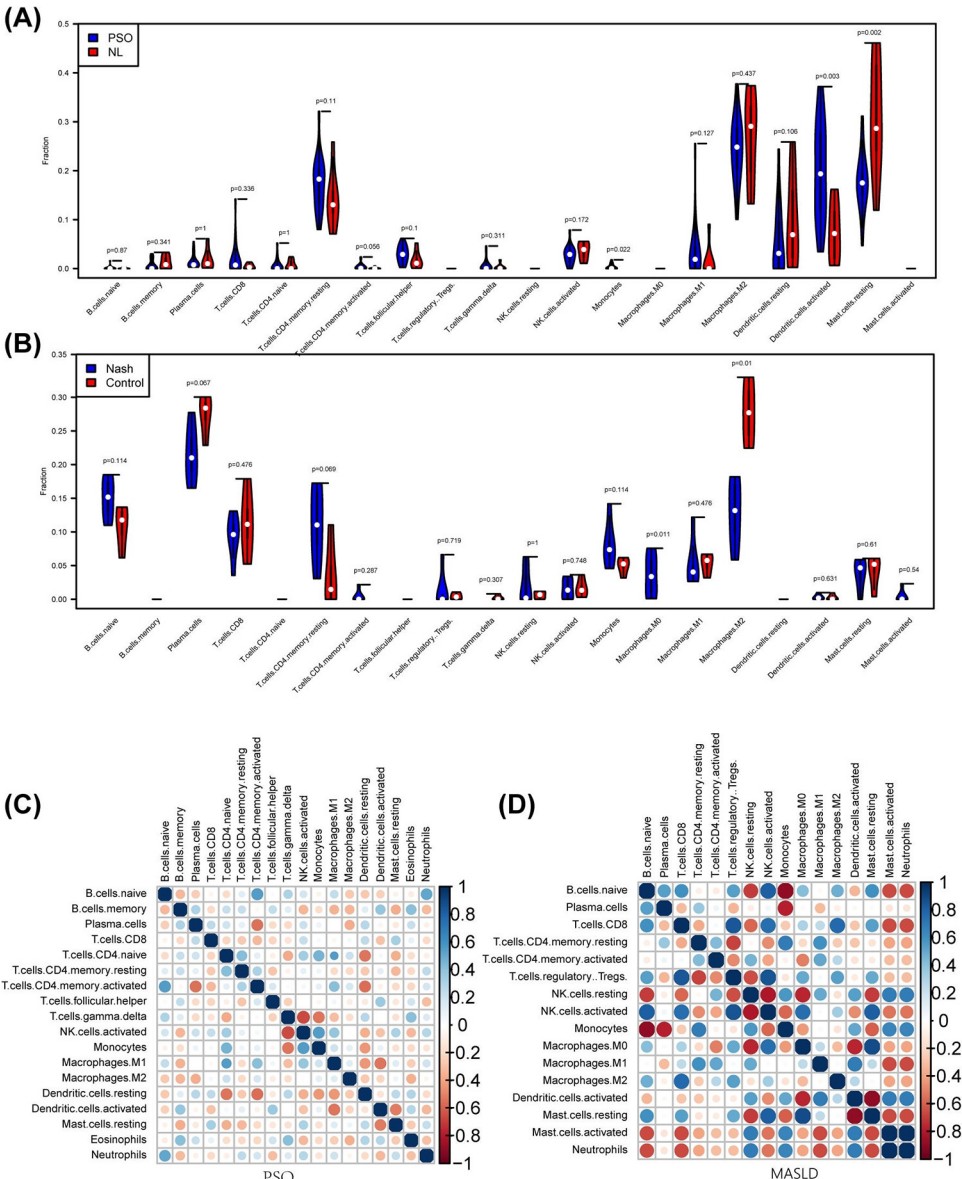

**Fig 7. Visualization of immune infiltration analysis.** (A) (B) Violin plots of immune cell infiltration in the disease versus control groups in GSE30999 and GSE89632. (C) (D) Correlation matrix consisting of 22 immune cell subtypes in the two diseases. Both horizontal and vertical axes indicate immune cell subtypes. The size of the circles and the depth of the color indicate the strength of the correlation, with darker colors indicating stronger correlations, blue indicating positive correlations, and red indicating negative correlations.

of psoriasis pathogenesis [13]. IL-17 axis plays an important role in MASLD [14], also observed in our study.

Th17 cells are a functionally polarized subpopulation of effector CD4$^+$ T cells and are considered to be the main producers of IL-17A and IL-17F. However, various immune cells (e.g., γδ T cells, CD8$^+$ T cells, NK cells, NKT cells, neutrophils, macrophages, dendritic cells, lymphoid tissue inducers, and intrinsic lymphocytes) and non-immune cells (e.g., intestinal plate cells) also express IL-17A and IL-17F. The specificity of the biological effects of family members of IL-17, as with other cytokines, depends on their interactions with signaling receptor

complexes. IL-23 and IL-17A play a key role is chronic inflammation's initiation and mainte-nance mechanisms [15, 16]. IL-23 stimulates Th17 cell differentiation, activation, proliferation, and survival and promotes the production of effector cytokines such as IL-17A and IL-22, IL-17 production is also independent of IL-23 [17–19]. A receptor for IL-17A, IL-17RA is the best-described receptor in the IL-17 family and is widely expressed in skin and liver. The main targets of IL-17 in psoriasis include keratin-forming cells, endothelial cells, and natural immune cells [20]. The liver consists mainly of hepatocytes, Kupffer cells, hepatic stellate cells, bile duct epithelial cells, and hepatic sinusoidal endothelial cells [14, 21, 22]. Activation of the IL-17 axis leads to the production of pro-inflammatory cytokines and chemokines that mobi-lize neutrophils [23], such as IL-8, IL-1β, IL-6 and CXCL1, CXCL2, CXCL3, CXCL5, and CXCL6 [24]. Similarly, IL-17A has a pro-inflammatory effect on antigen-presenting cells (including macrophages) [25], constantly triggering inflammation, and creating a vicious cycle. Gene function enrichment and pathway enrichment analysis identified differential genes common to both diseases that function primarily in IL-17 related pathways, suggesting that these genes may be central to both diseases. In the presence of these genes, the two dis-eases exhibit certain similar features, and in a deeper pathological mechanism, the two diseases contribute to each other's influence, as we have observed in the clinic.

Our study also found that activation of the cGAS-STING pathway was also found to play an important role in both diseases in our study. The basic function of the cGAS-STING pathway has the essential function of detecting and limiting the spread of exogenous DNA. The most prominent function of this pathway in response to microbial nucleic acids is the production of inflammatory and antiviral proteins, mainly type I interferon (IFN-I)2 [26–28]. IFNβ can be produced by almost any type of cell in response to nucleic acid stimulation [29]. The release of this cytokine can initiate or amplify IFN-I production in other cells, especially plasmacytoid dendritic cells [30]. IFN-I is highly associated with autoimmunity and is highly immunostimu-latory, inducing dendritic cells activation and maturation, upregulating expression of MHC and regulatory molecules, activating NK cells, T cells and B cells, and suppressing regulatory T cells [31]. The activation of STING also leads to the production of other inflammatory cyto-kines, such as TNF, IL-6 and IL-1β, which have a very broad activation potential both inside and outside the immune system [32]. However, the binding of DNA to cGAS is not sequence-specific [33], and in addition to reacting to DNA of pathogenic origin, this pathway can also produce an inflammatory response to its nucleic acids, which are more likely to cause an auto-immune inflammatory response [32]. We found that in the pathogenesis of psoriasis and MASLD, this inflammation caused by microbial or autochthonous factors is significantly dif-ferent for normal levels. Comparing single-cell datasets from both diseases, we also found that the corresponding DCs and macrophages may be the key cells activating this pathway. DNA from own dead cells and external microbial nucleic acids leads to increased STING and inflammatory responses, causing a range of responses such as dipose inflammation and insulin resistance in vivo [11, 29, 34, 35]. Could we treat both diseases by inhibiting the activation of the cGAS-STING pathway, especially in patients with psoriatic relapses with or without the conversion of steatohepatitis to nonalcoholic steatohepatitis?

Comparing single-cell data and immune infiltration in both diseases, we found that CD4[+] T cells, myeloid cells such as macrophages (Kupffer cells), DCs, or granulocytes were signifi-cantly higher in the disease group than in the control group. Classical T lymphocytes (CD4[+] or CD8[+]) play a central role in maintaining liver immunity [36], and imbalances in the lympho-cyte population contribute to inflammation in patients. Also, cytokines secreted by CD4[+] T cells such as (Th1, Th2, Th17) and γδ T cells exacerbate the inflammatory response of patients. A similar infiltration of lymphocyte populations is present in psoriatic lesions. In the myeloid cell population, DCs and macrophages provide an abundance of IL-23 [37], which is required

to expand and survive IL-17-producing T cells. Macrophages in the liver are derived from circulating monocytes recruited by resident Kupffer cells, and these macrophages can be classified into M0, M1, and M2 types. The type M1 triggers inflammation and fibrosis in MASLD, and one of the most important cytokines from Kupffer cells and monocyte-derived hepatic macrophages that function TNF, which plays a role in steatosis in MASLD degeneration, inflammation and development of fibrosis [38]. These circulating monocytes have more pro-inflammatory than the resident Kupffer cells [39]. As we have observed, both patients with psoriasis and patients with MASLD contain high numbers CD163+ macrophages, and the inflammatory response in patients with psoriasis leads to the recruitment of monocytes to the liver, exacerbating the hepatic inflammatory response and increases the incidence of MASLD and the probability of progression to NASH with increased fibrosis, and vice versa. NK cells may contribute to the development of fibrosis by releasing IFN- γ, TNF, and IL-22, among other cytokines, in psoriasis and MASLD [40]. NK cells are the main producers of hepatic IFN-γ. Also, they produce IL-17A, which further promotes inflammation and is involved in liver injury, fibrosis and regeneration [36, 41, 42], with elevated expression in both diseases, along with complex macrophage and CD4[+] T cell Cell-Cell communication, which may be related to the maintenance of inflammation in both diseases.

The two main target cell types of psoriasis, keratin-forming cells and fibroblasts, both have insulin receptors and insulin-like growth factor receptors [43]. Insulin has been reported to cross the dermal-epidermal junction and affect keratin-forming cells, and the downstream PI3-K/Akt/mTOR signaling cascade is a common and important signaling pathway in obesity, insulin resistance and psoriasis [44]. The mixture of these inflammatory factors also disrupts the normal insulin-associated PI3-K/AKT pathway, causing insulin resistance [45]. Conversely, similar lipid abnormalities, including hypertriglyceridemia, low plasma HDL concentrations, and massive accumulation of VLDL, were shown in animal models of insulin resistance and human patients. Insulin resistance leads to increased secretion of very low-density lipoproteins, resulting in increased transport of fatty acids to the liver. With the increased transport of lipids to the hepatocytes, large amounts of apolipoprotein B (ApoB) are synthesized and accumulate in the endoplasmic reticulum, which can lead to chronic metabolic imbalances that alter the metabolic homeostasis of the endoplasmic reticulum and impair its function [46]. Stress in the endoplasmic reticulum can further activate the inflammatory kinases JNK and NF-kB, leading to metabolic inflammation in the liver and activation of insulin resistance [47, 48]. Most of the functions of the genetic association network we have constructed are also mostly involved in chronic systemic inflammation caused by lipotoxicity.

Psoriasis and MASLD share many of the same processes in their pathogenesis, especially the development of chronic inflammation, which is complex, involves multiple cells and systems and requires more and more in-depth studies. Pathways associated with IL-17 are central to both diseases, and we have also identified key genes and transcription factors that regulate both. In particular, the cGAS-STING pathway and specific inflammatory macrophages may play key roles in the induction phase of inflammation. All these need to be further investigated.

## 5. Conclusion

Our study reveals the common pathogenesis of psoriasis and MASLD from gene expression to immune cell similarities and differences, identifies key genes and regulatory pathways common to both, and elucidates the similarities in the immune microenvironment of both diseases, providing new ideas for subsequent studies on targeted therapy.

## Supporting information

**S1 Table. Pathway gene set.**
(XLSX)

## Acknowledgments

We thank Biomamba for their guidance in bioinformatics and data analysis for the current study.

## Author Contributions

**Conceptualization:** Shuhui Tan, Mingyue Liu.

**Data curation:** Shuhui Tan, Mingyue Liu.

**Formal analysis:** Shuhui Tan, Mingyue Liu.

**Investigation:** Shuhui Tan, Mingyue Liu.

**Methodology:** Fei Feng, Zhenhua Nie.

**Supervision:** Zhenhua Nie.

**Validation:** Shuhui Tan, Mingyue Liu, Rui Tian.

**Visualization:** Shuhui Tan, Mingyue Liu.

**Writing – original draft:** Shuhui Tan, Mingyue Liu, Fei Feng, Ruicheng Li.

**Writing – review & editing:** Shuhui Tan, Mingyue Liu, Fei Feng, Ruicheng Li, Rui Tian.

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
