## [Decision Letter · Decision Letter 0]

4 Apr 2024

PONE-D-23-37934Exploring the Pathogenesis and Immunological Profiles of Psoriasis Complicated with Non-alcoholic Fatty LiverPLOS ONE

Dear Dr. Feng,

Thank you for submitting your manuscript to PLOS ONE. After careful consideration, we feel that it has merit but does not fully meet PLOS ONE’s publication criteria as it currently stands. Therefore, we invite you to submit a revised version of the manuscript that addresses the points raised during the review process.

We look forward to receiving your revised manuscript.

Kind regards,

Seth Agyei Domfeh, PhD

Academic Editor

PLOS ONE

Journal Requirements:

3. Please remove your figures from within your manuscript file, leaving only the individual TIFF/EPS image files, uploaded separately. These will be automatically included in the reviewers’ PDF.

Reviewers' comments:

Reviewer's Responses to Questions

**Comments to the Author**

1. Is the manuscript technically sound, and do the data support the conclusions?

Reviewer #1: Yes

Reviewer #2: Partly

2. Has the statistical analysis been performed appropriately and rigorously? 

Reviewer #1: Yes

Reviewer #2: Yes

3. Have the authors made all data underlying the findings in their manuscript fully available?

Reviewer #1: Yes

Reviewer #2: Yes

4. Is the manuscript presented in an intelligible fashion and written in standard English?

Reviewer #1: Yes

Reviewer #2: Yes

5. Review Comments to the Author

Reviewer #1: The study by Fei Feng et al is quite interesting addressing a hot topic, the interaction between psoriasis and Metabolic-associated steatotic liver disease (MASLD).

MAJOR COMMENTS:

1.- I suggest shifting nomenclature from NAFLD to the recent consensus MASLD across the manuscript.

2.- The study is quite interesting well written and addressed from a “trans-omic” perspective supporting an intriguing overlap in the pathophysiology of Psoriasis and MASLD that could impact on therapeutic management.

3.- It could be interesting adding data from clinical trials using anti-IL17 drugs to treat psoriasis on liver disease improvement. Data are scarce and authors emerged some questions over the discussion. I suggest reviewing available clinical data.

Reviewer #2: Lines 38-39 autoimmune pathogenic characteristics – there is actually an immune pathogenic, not autoimmune, since autoimmunity is not involved in psoriasis.

Lines 40-42 that presentation is over simplified.

Lines 55-56 there are several articles published concerning the common pathogenesis between these psoriasis and NAFLD, that should probably be cited here, one of the most recent being here – https://doi.org/10.3390/ijms25052660

Line 133 infiltration. of immune cells This – please correct the position of the dot

Lines 265-267 please reformulate as is rather confusing

Also, https://doi.org/10.3390/ijms23095198 might be worth cited as it describes the roles of inflammation in psoriasis and associated disease.

6. PLOS authors have the option to publish the peer review history of their article (what does this mean?). If published, this will include your full peer review and any attached files.

Reviewer #1: **Yes: **MANUEL ROMERO-GOMEZ

Reviewer #2: **Yes: **Daniel O. Costache

---

## [Author Response · Author response to Decision Letter 0]

14 May 2024

Dear Editors and Reviewers:

Thank you for your feedback on our article.These comments have important guiding and research significance for the revision and improvement of the paper. We have carefully studied the comments and made corrections, hoping for approval. The specific modifications are as follows:

Reviewer#1: 

1.Response to comment:  I suggest shifting nomenclature from NAFLD to the recent consensus MASLD across the manuscript.

Response: We have read the latest references and replaced "non-alcoholic fatty liver disease (NAFLD)" with " metabolic dysfunction-associated steatotic liver disease (MASLD)" in the text. And the definition of MASLD has been rephrased.

2.Response to comment: The study is quite interesting well written and addressed from a “trans-omic” perspective supporting an intriguing overlap in the pathophysiology of Psoriasis and MASLD that could impact on therapeutic management.

Response: Thank you for your reply. We hope our article can provide some reference for treatment management.

3.Response to comment: It could be interesting adding data from clinical trials using anti-IL17 drugs to treat psoriasis on liver disease improvement. Data are scarce and authors emerged some questions over the discussion. I suggest reviewing available clinical data.

Response: Sorry, we have screened and analyzed liver disease patients who are currently using anti IL-17 drugs to treat psoriasis in clinical practice, but have not obtained sufficient case data. We will continue to collect and analyze data in this area in the future.

Reviewer#2: 

1.Response to comment: Lines 38-39 autoimmune pathogenic characteristics – there is actually an immune pathogenic, not autoimmune, since autoimmunity is not involved in psoriasis.

Response: We will replace“ The skin disease, also known as psoriasis, has a strong genetic predisposition, and autoimmune pathogenic characteristics” with “The skin disease，also known as psoriasis, has a strong genetic predisposition and immune-related pathogenic characteristics.”

2. Response to comment： Lines 40-42 that presentation is over simplified.

Response: We have added a description of the clinical symptoms and site of onset of psoriasis. We will replace“Psoriasis plaque is characterized by erythematous and scaly red or salmon-pink lesions, often covered by white or silvery plaques. Generally, the scalp, elbows, knees and lower back are the preferred sites, with a bilateral symmetrical distribution. Patients also suffer from significant psychological stress, which can greatly affect their quality of life” with “Psoriasis lesions are well-demarcated round red papules or plaques with dry gray or silvery-white scales produced by hyperkeratosis of the skin, and in some subtypes, pustules. The lesions are usually symmetrically distributed on the scalp, elbows, knees, lumbosacral area and body folds, and may also involve the oral mucosa, tongue or eyelids. Psoriasis reduces the patient's quality of life, increases financial burden, causes psychological stress, and in severe cases, can be combined with other life-threatening complications.”

3. Response to comment：Lines 55-56 there are several articles published concerning the common pathogenesis between these psoriasis and NAFLD, that should probably be cited here, one of the most recent being here – https://doi.org/10.3390/ijms25052660

Response: We have read the article and rephrased the public pathogenesis of psoriasis and MASLD. Specifically，We will replace“In recent years, the prevalence of NAFLD has gradually increased. Some observational studies have shown that psoriasis increases the prevalence of NAFLD, but the common pathogenesis between these two diseases has never been fully identified. Insulin resistance and inflammatory cytokine accumulation are thought to play an important role in the mechanism of association between the two diseases.”with“ Although some studies have shown that psoriasis increases the prevalence of MASLD , the common pathogenesis between these two diseases has been a point of wide interest and discussion. It is now believed that obesity, insulin resistance, persistent inflammation, and the hepatic-dermal axis that develops between the skin, adipose tissue, and liver play important roles in the mechanism of association between the two diseases.” And cited this article.

3. Response to comment：Line 133 infiltration. of immune cells This – please correct the position of the dot.

Response: We have made corrections to the position of the dot.

4. Response to comment：Lines 265-267 please reformulate as is rather confusing

Also, https://doi.org/10.3390/ijms23095198 might be worth cited as it describes the roles of inflammation in psoriasis and associated disease.

Response: We have rephrased it as follows:“the point of action may arise from the indirect adaptive immune effects of the IL-23 /IL-17A axis” has been replaced with " the IL-23/IL-17A axis at the core of psoriasis pathogenesi.” And cited "kin Inflammation Modulation via TNF-α, IL-17, and IL-12 Family Inhibitors Therapy and Cancer Control in Patients with Psoriasis" as a supplementary argument for the article.

We have carefully addressed the reviewers' suggestions, offering a point-to-point response to their comments. Enclosed, please find the revised manuscript with changes highlighted, along with a "clean" copy formatted according to the journal's requirements. We believe that these revisions, guided by the reviewers' critiques and Manuscript Edits, have substantially enhanced the quality of the manuscript, and we hope it meets the standards for publication in PLOS ONE.

We appreciate your consideration in allowing us to revise our manuscript and eagerly anticipate your feedback.

Sincerely,

Shuhui Tan,Mingyue Liu

---

## [Decision Letter · Decision Letter 1]

27 May 2024

Exploring the Pathogenesis and Immunological Profiles of Psoriasis Complicated with MASLD

PONE-D-23-37934R1

Dear Dr. Feng,

We’re pleased to inform you that your manuscript has been judged scientifically suitable for publication and will be formally accepted for publication once it meets all outstanding technical requirements.

Kind regards,

Seth Agyei Domfeh, PhD

Academic Editor

PLOS ONE

---

## [Editor Report · Acceptance letter]

14 Jun 2024

PONE-D-23-37934R1 

PLOS ONE

Dear Dr. Feng, 

I'm pleased to inform you that your manuscript has been deemed suitable for publication in PLOS ONE. Congratulations! Your manuscript is now being handed over to our production team.

Kind regards, 

on behalf of

Dr. Seth Agyei Domfeh 

Academic Editor

PLOS ONE